# Wood xerogel for fabrication of high-performance transparent wood

**Shennan Wang[1], Lengwan Li[2], Li Zha[1], Salla Koskela [1,2], Lars A. Berglund [2] & Qi Zhou [1,2]** ✉

Optically transparent wood has been fabricated by structure-retaining delignification of wood and subsequent infiltration of thermo- or photocurable polymer resins but still limited by the intrinsic low mesopore volume of the delignified wood. Here we report a facile approach to fabricate strong transparent wood composites using the wood xerogel which allows solvent-free infiltration of resin monomers into the wood cell wall under ambient conditions. The wood xerogel with high specific surface area (260 $m^2\,g^{-1}$) and high mesopore volume (0.37 $cm^3\,g^{-1}$) is prepared by evaporative drying of delignified wood comprising fibrillated cell walls at ambient pressure. The mesoporous wood xerogel is compressible in the transverse direction and provides precise control of the microstructure, wood volume fraction, and mechanical properties for the transparent wood composites without compromising the optical transmittance. Transparent wood composites of large size and high wood volume fraction (50%) are successfully prepared, demonstrating potential scalability of the method.

Biocomposites reinforced with renewable natural fibers or fillers have attracted major attention on their merits of both high performance and favorable sustainability[1]. In particular, the fabrication of transparent wood has revolutionized the material design strategy by employing the top-down structure retaining delignification of wood to overcome the challenging bottom-up assembly of natural fibers and nanocellulose into a hierarchical structure in a polymer matrix[2–4]. The voids in delignified wood have been filled with refractive index-matched polymers to produce transparent composites with low density, high mechanical performance, and low thermal conductivity, which are the essential features of nanocellulose-based composites. In addition, the delignification process preserves the unique hierarchical structure of wood and the native alignment of cellulose microfibrils, enabling strong micro/nanoscale reinforcement. Thus, the obtained transparent wood composites showed unique anisotropic behavior in light transmission[5], heat transfer[6], and fracture behavior[3], which are advantageous in building[7,8], photonic[9], or optoelectrical materials[10] applications. Further chemical modification of delignified wood by

surface acetylation[11] or elimination of excessive lignin[12] allowed the preparation of thick yet transparent wood composites. However, limited by the low mesopore volume and low permeability of cell wall in delignified wood, the successful impregnation of monomers or prepolymers still requires the use of solvent and vacuum[2,11,12]. By contrast, highly mesoporous nanocellulose networks prepared by bottom-up approach, such as nanopapers and aerogels, allow solvent-free and homogeneous impregnation of matrix polymers owing to the high specific BET (Brunauer, Emmett, and Teller) surface area ($S_{BET}$) (>350 $m^2\,g^{-1}$), and the strong capillary force generated by mesopores with sizes in the range of 2–50 nm[13,14].

Complete or partial removal of lignin usually generates mesopores and endows delignified wood with $S_{BET}$ in the range of 9–41 $m^2\,g^{-1}$ depending on the drying methods[15]. The delignification of wood is generally achieved by either pulping process, or bleaching process, or a combination of both[4]. Nanowood was produced by the alkaline sulfite pulping process using $NaOH/Na_2SO_3$, followed by an additional bleaching process with $H_2O_2$[16]. Such delignified wood had a

[1]Division of Glycoscience, Department of Chemistry, School of Engineering Sciences in Chemistry, Biotechnology and Health, KTH Royal Institute of Technology, AlbaNova University Centre, Stockholm SE-106 91, Sweden. [2]Wallenberg Wood Science Center, Department of Fibre and Polymer Technology, KTH Royal Institute of Technology, Stockholm SE-100 44, Sweden. ✉e-mail: qi@kth.se

$S_{BET}$ of 13.8 $m^2 g^{-1}$ after freeze drying[17]. Wood sponge prepared by the bleaching process with a mixture of $H_2O_2$ and acetic acid had a $S_{BET}$ of 20 $m^2 g^{-1}$ after freeze drying[15,18]. Delignified wood aerogel with a $S_{BET}$ of 36 $m^2 g^{-1}$ was also produced by the bleaching process with acidified sodium chlorite followed by supercritical drying with $CO_2$[19]. Further increase of mesopores inside the cell wall requires in situ fibrillation of cellulose microfibril bundles to create interfibrillar spacing that falls in the mesopore range[20,21]. Individualization of cellulose microfibrils is often facilitated by chemical pretreatment, such as 2,2,6,6-tetra-methylpiperidine-1-oxyl radical (TEMPO)-mediated oxidation[22,23]. By applying TEMPO-mediated oxidation to delignified wood, successful in situ fibrillation of cellulose microfibrils in the wood cell wall was achieved without impairing the native hierarchical structure of wood and alignment of cellulose microfibrils. The wood aerogel with a $S_{BET}$ of 249 $m^2 g^{-1}$ was then produced by supercritical drying with $CO_2$[19]. Alternatively, the creation of a mesoporous cellulose network in delignified wood was obtained by ionic liquid-assisted dissolution-regeneration of cellulose nanofibers in the cell lumina, yielding a high $S_{BET}$ of 181 $m^2 g^{-1}$ after supercritical drying with $CO_2$[24]. The aggregation of cellulose microfibrils in the cell wall induced by ice crystal growth was inevitable during the freeze-drying process. Solvent exchange coupled with supercritical drying was essential in preserving the nanofiber network structure[25]. However, the cell wall strength and modulus of the supercritical dried wood aerogel were significantly lower than that from freeze-dried wood aerogel due to reduced interfibrillar bonding between cellulose microfibrils caused by solvent exchange from water to organic solvent[19]. Therefore, crosslinking between the cellulose microfibrils before the solvent exchange step is necessary to enhance the mechanical performance.

Herein, we report a highly feasible approach to obtain wood xerogels for fabrication of high-performance transparent wood with high wood volume fraction. Delignified wood (D-wood) was oxidized by TEMPO-mediated oxidation followed by ionic crosslinking with trivalent aluminum ions ($Al^{3+}$). Instead of freeze-drying or supercritical drying, the obtained TEMPO-oxidized wood (TO-wood) was solvent exchanged with hexane and dried at 60 °C under ambient pressure to produce a xerogel, maintaining high specific surface area and high mesopore volume in the cell wall. The highly mesoporous structure of TO-wood xerogel allowed solvent-free impregnation of UV curable acrylic resin ethoxylated bisphenol A diacrylate (ABPE-10) in the wood cell wall under ambient conditions, producing highly transparent wood (Fig. 1a). The wood composite prepared from the non-oxidized D-wood xerogel and ABPE-10 was opaque. In addition, the TO-wood xerogel was compressible (Fig. 1b) and enabled the preparation of transparent wood with excellent mechanical performance and high optical transparency with a wood volume fraction as high as 50% and a thickness of 6 mm, comparable to the standard 1/4 inch single-pane window glass. The effect of high cell wall mesopore volume on the cell wall-polymer matrix interaction and interface structure in the wood xerogel-based transparent wood as well as the relation of the structure to the optical and mechanical properties were studied.

## Results

### Fabrication of wood xerogels
Balsa wood with a density of 350 $kg m^{-3}$ was delignified in acidified sodium chlorite solution and further oxidized using the TEMPO-mediated oxidation method to produce TO-wood[19]. The never dried TO-wood was further ionically crosslinked with $Al^{3+}$. Oven drying of delignified wood was reported to induce irreversible collapse of the cell wall cavity, which led to reduced accessibility by water vapor[26]. Solvent exchange to organic solvents, e.g. acetone[27], octane[28], hexane[29], with low surface tension has been employed to maintain the fibrillated structure in the cellulose nanofibril network and to reduce the capillary force that induces nanoscale shrinkage. Therefore, monolithic xerogel of TO-wood with a white appearance was

subsequently obtained after evaporative drying from hexane in an oven at 60 °C. The aluminum content in $Al^{3+}$-crosslinked TO-wood xerogel was 0.8 wt.%. The successful crosslinking of TO-wood xerogel with $Al^{3+}$ was confirmed by FTIR, as the carboxylic group at 1730 $cm^{-1}$ in the spectrum of non-crosslinked TO-wood xerogel disappeared (Supplementary Fig. 1). The non-crosslinked sample showed tensile strength of 0.80 MPa, Young's modulus of 0.11 GPa, and strain-to-failure of 1.36% along the longitudinal direction (Supplementary Fig. 2). After crosslinking with $Al^{3+}$ ions, the mechanical properties of TO-wood xerogel were significantly improved, showing tensile strength of 4.81 MPa, Young's modulus of 0.32 GPa, and strain-to-failure of 3.62%. This indicates that the crosslinking with $Al^{3+}$ ions reduced the structural defects in the wood xerogel through enhanced interfibrillar bonding of cellulose microfibrils.

### Structure of wood xerogels
Field emission scanning electron microscopy (FE-SEM) analysis (Fig. 2a) revealed that the D-wood xerogel showed a well-preserved cellular structure. The intercellular space, including the middle lamella and cell corner, remained intact and unopen after removal of lignin. The secondary cell wall, which typically accounts for 70–90% of the wall mass, showed a compact structure with aligned cellulose microfibril bundles. At the cell wall surface perpendicular to the fibers, dense packing of cellulose microfibrils was observed without detectable voids. By contrast, the TO-wood xerogel showed a cellular but opened structure due to extensive removal of lignin and hemicelluloses in the middle lamella and cell corner. Cellulose microfibril bundles were fibrillated by TEMPO-mediated oxidation, generating nanoscale voids, as observed on the cross section of the secondary cell wall. The cell wall surface of the TO-wood xerogel showed a highly fibrillated structure composed of individualized cellulose microfibrils and nanoscale pores smaller than 100 nm. The mesoporous structure of the TO-wood xerogel corresponded to a higher porosity of 88% than that (84%) for the D-wood xerogel (Supplementary Table 1).

The nanostructural difference between the TO-wood and D-wood xerogels was also characterized by $N_2$ physisorption. Both xerogels presented type IV physisorption isotherms and H3 hysteresis loops (Fig. 2b), which suggest a mesoporous structure with the presence of macropores larger than 100 nm. The TO-wood xerogel showed 5 times higher adsorbed $N_2$ than the D-wood xerogel at relative pressure ($p/p_0$) of 0.98, implying a substantially larger porosity and surface area. The $S_{BET}$ was as high as 260 $m^2 g^{-1}$ for the TO-wood xerogel, while a much lower $S_{BET}$ of 37 $m^2 g^{-1}$ was recorded for the D-wood xerogel. This is a significant improvement over the conventional foam- and network-type xerogels[27,28]. The pore size distribution was derived from the adsorption isotherm that was fitted with the $N_2$-DFT model based on the non-local density function theory assuming a slit-like pore geometry. Interestingly, in both xerogels, the size of mesopores fell in a narrow range of 2–20 nm (Fig. 2c). Previous results on the supercritical dried TO-wood aerogel showed mesopores mainly over the range of 10–50 nm[19]. The current result suggests that the xerogel process can effectively minimize the drying-induced elimination of small interfibrillar spacing between cellulose microfibrils, thus preventing the aggregation of cellulose microfibrils during drying. The preservation of the nanostructure was also partially affected by ionic crosslinking and the wood density. The $S_{BET}$ values for $Al^{3+}$ crosslinked xerogels were higher than the corresponding non-crosslinked ones (Supplementary Fig. 3 and Table 1). The TO-wood xerogels obtained from balsa wood with a lower density of 190 and 280 $kg m^{-3}$ as the starting material had a lower $S_{BET}$ of 130 and 155 $m^2 g^{-1}$, respectively. This was anticipated since a higher concentration of cellulose microfibrils in the cell wall of high-density balsa wood reduced the shrinkage of the fibrillated cell wall structure during drying[25]. In particular, the TO-wood xerogel from high-density balsa wood showed the highest mesopore volume of 0.37 $cm^3 g^{-1}$ (Supplementary Table 1), more than 5

times larger than the D-wood xerogel (0.07 cm$^3$ g$^{-1}$). It was hypothesized that a higher mesopore volume would enhance cell wall accessibility and permeability and thus benefit the infiltration of polymers/monomers and the fabrication of high-performance transparent wood[30].

### Fabrication of transparent wood

Both TO-wood and D-wood xerogels were impregnated in the UV curable acrylic resin ABPE with a refractive index of 1.536 which has been previously used to fabricate transparent nanocellulose composites for optoelectrical applications[31]. The impregnation process was performed under ambient conditions without the use of solvent or vacuum. Owing to high mesopore volume and strong capillary force generated in the slit-like pore geometry[32], a faster infiltration of ABPE

resin was observed for the TO-wood xerogel, which was completed after 12 hours as indicated by the transformation of the xerogel from optically white to transparent (Supplementary Fig. 4). The D-wood xerogel immersed in ABPE resin remained white and opaque, suggesting incomplete impregnation into large aggregates of cellulose microfibrils that extensively reflect incident light[30]. The ABPE acrylic resin-impregnated xerogels were cured with UV light (365 nm) for 15 min to produce transparent wood. The fast infiltration of the resin in TO-wood xerogel was also confirmed by contact angle measurement (Supplementary Fig. 5). The ABPE resin had an initial contact angle of 73.8° on the TO-wood xerogel surface and the droplet was quickly absorbed into the wood within 5 s. By contrast, the resin droplet on D-wood xerogel had an initial contact angle of 92.5° which decreased to 60.8° at 5 s and remained at 58.4° after 10 minutes.

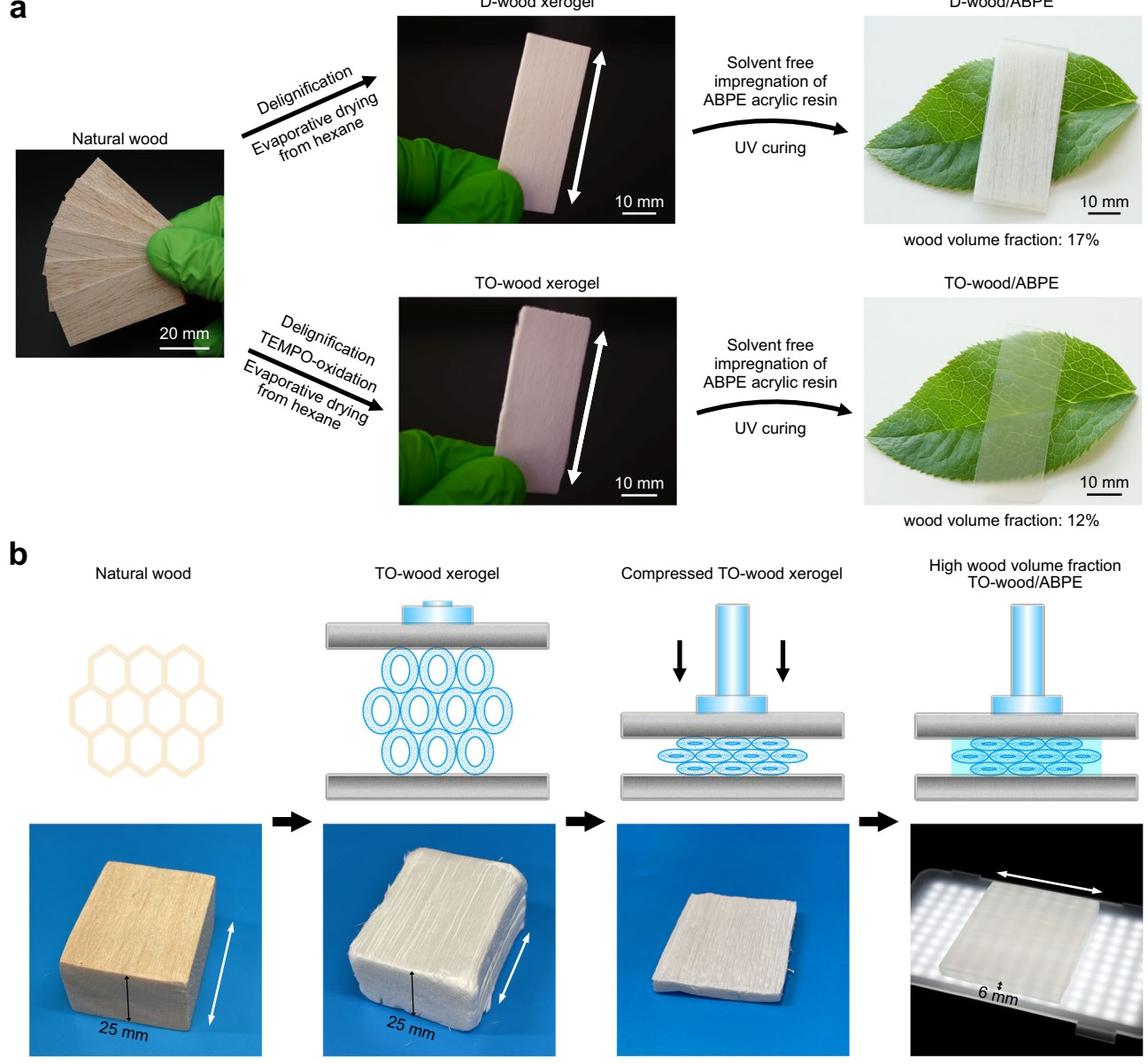

**Fig. 1 | Fabrication of wood xerogels and transparent wood. a** Photographs showing the white D-wood/ABPE composite (wood volume fraction of 17%) and the transparent TO-wood/ABPE composite (wood volume fraction of 12%) prepared from the corresponding D-wood and TO-wood xerogels with white arrows indicating the axial direction of fiber cells as in the native balsa wood. **b** Schematic illustration for compression of the TO-wood xerogel to prepare thick transparent wood with a wood volume fraction of 50% and corresponding photographs of the samples.

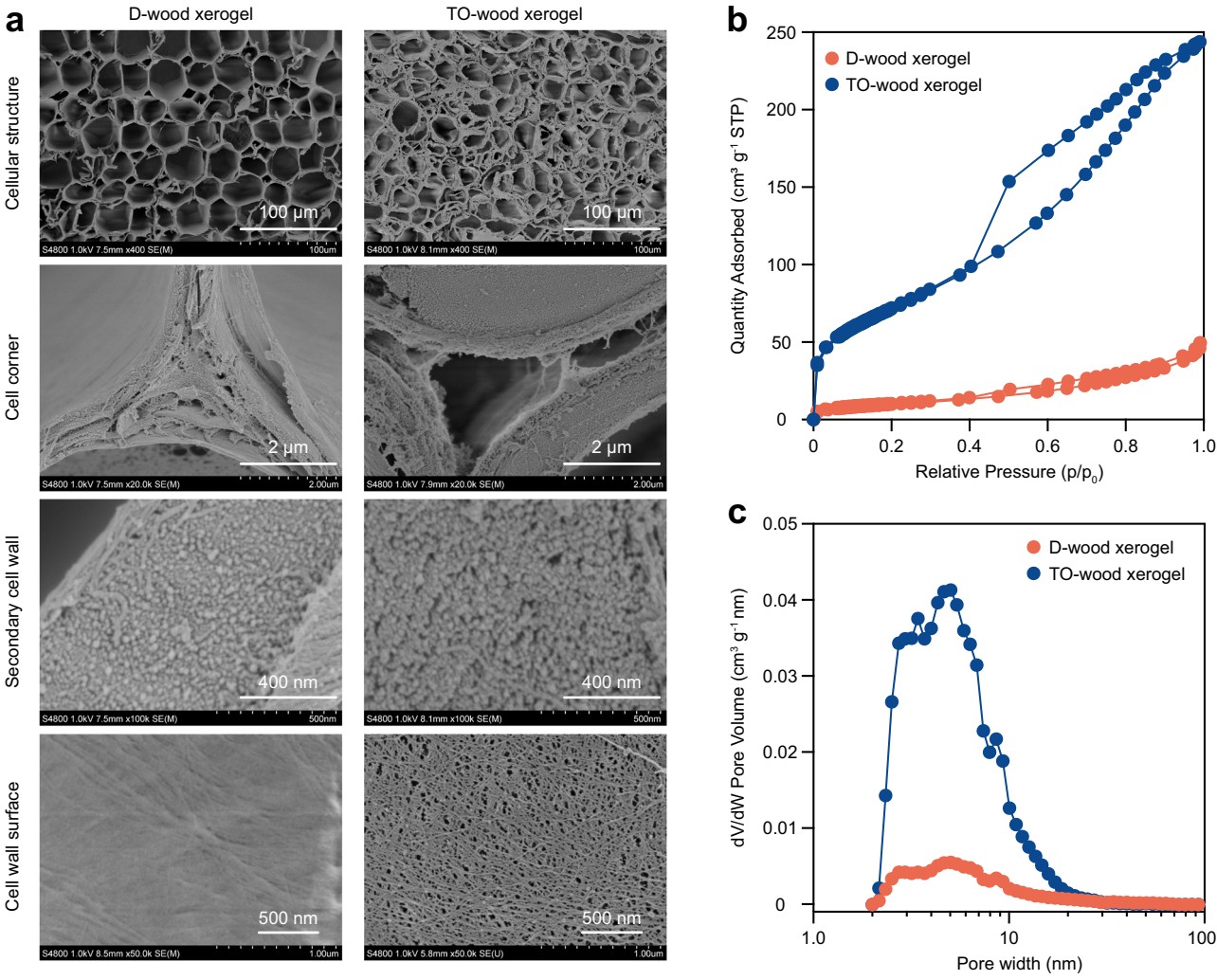

**Fig. 2 | Structural characterization of wood xerogels. a** FE-SEM micrographs of the D-wood and TO-wood xerogels, showing the cellular structure, cell corner, secondary cell wall, and cell wall surface, respectively. **b** Nitrogen adsorption-desorption isotherms of the TO-wood xerogel and the corresponding D-wood xerogel. **c** Pore size distribution derived from the $N_2$ adsorption isotherm using a method based on the non-local density functional theory.

## Structure and optical properties of transparent wood

The TO-wood/ABPE composite with a wood volume fraction of 12% showed a total optical transmittance at 550 nm ($T_{550nm}$) of 81.1%, slightly lower than that of the neat acrylic resin sheet (90.1%) (Fig. 3a). By contrast, the D-wood/APBE composite with a wood volume fraction of 17% showed an opaque appearance with a $T_{550nm}$ value of 38.7%. The microstructure difference also strongly affected the scattering behavior of light through transparent wood. The D-wood/ABPE composite showed a high haze of 95.5%, indicating a strong forward light scattering effect through the sample. Individualization of cellulose microfibrils effectively reduced light scattering and resulted in a significantly reduced haze in the TO-wood/ABPE composite. The intrinsic anisotropy of the wood structure was well maintained in the TO-wood/ABPE composite. The light scattering pattern of the TO-wood/ABPE composite showed discrete scattering angles between directions along and perpendicular to the fiber direction, which was not observed for the neat ABPE acrylic resin sheet (Supplementary Fig. 6). These distinct optical properties are partially derived from the interfacial interaction between the cell wall and polymer matrix[33]. Interfacial gaps between the cell wall and acrylic resin inside the cell lumen were observed on the microtome-trimmed cross section of the D-wood/ABPE composite with FE-SEM (Fig. 3b). Poor penetration of the acrylic resin into the cell wall led to a clear phase separation. With much increased cell wall

mesopore volume, a good cell wall-acrylic resin interface was formed in the TO-wood/ABPE composite. The ABPE acrylic resin was also found in the middle lamella and cell wall corner regions, where the previously formed empty intercellular space was filled with acrylic resin to maintain structural integrity.

The 2D small angle X-ray scattering (SAXS) results revealed the interfacial structure between the acrylic resin polymer and wood cell wall. The plot of scattering intensity $I(q)$ as a function of the scattering vector $q$ (Fig. 3d) was extracted from the 2D SAXS patterns (Fig. 3c). The $I(q)$ curve of the D-wood/ABPE composite showed a broad shoulder centered at $q = 0.715\ \text{nm}^{-1}$, which corresponded to a length scale of 8.8 nm by employing Bragg's law ($d = 2\pi/q$)[34]. In the $I(q)$ curve of the TO-wood/ABPE composite, no shoulder peak was detected. One possible explanation arises from the relatively low contrast between the wood cell wall and acrylic resin polymer matrix. The calculated X-ray scattering length density (SLD) for ABPE acrylic resin was $11.009 \times 10^{-6}\ \text{Å}^{-2}$, which is close to cellulose with an SLD of $13.561 \times 10^{-6}\ \text{Å}^{-2}$. When the penetration of acrylic resin polymer into the cell wall was sufficient and the presence of air-filled gaps was minimized, the scattering feature in the $I(q)$ curve was less apparent[2,34]. This was also supported by the low porosity (0.6%) for the TO-wood/ABPE composite (Supplementary Table 2). The existence of spacing between scattering domains in the D-wood/ABPE composite suggests the

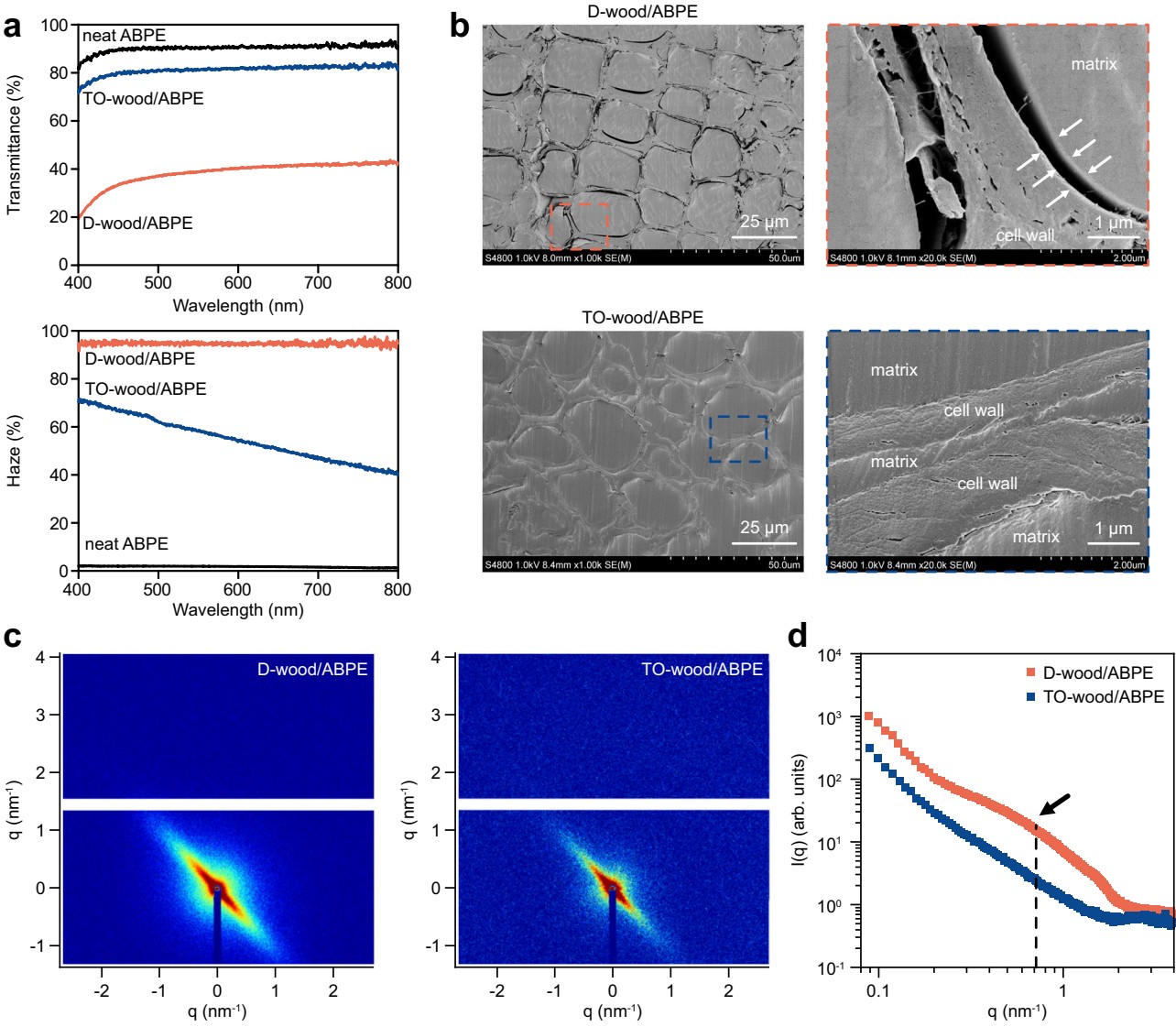

**Fig. 3 | Optical properties and structural characterization of transparent wood. a** Total light transmittance and haze of the neat ABPE acrylic resin sheet, TO-wood/ABPE and D-wood/ABPE composites (thickness: ca. 1.10 mm). **b** FE-SEM micrographs of the cross sections of the D-wood/ABPE and TO-wood/ABPE composites. **c** Typical 2D SAXS patterns of the D-wood/ABPE and TO-wood/ABPE composites.

**d** Scattering intensity, $I(q)$, as a function of the scattering vector $q$ on a logarithmic scale extracted from the average of 2D patterns. The black arrow indicates the shoulder peak position. Photographs of the white D-wood/ABPE and transparent TO-wood/ABPE composite samples are shown in Fig. 1a.

presence of unfilled voids inside the cell wall or at the cell lumen surface, which was also supported by the extensive phase separations observed in FE-SEM micrographs (Fig. 3b) and a larger porosity of 6.3%. The loose cluster of cellulose microfibrils in TO-wood likely improved the penetration of acrylic resin into the mesoporous cell wall and contributed to the higher $T_{550nm}$ and reduced haze values for the TO-wood/ABPE composite.

### Effect of wood volume fraction on properties of transparent wood

The in situ individualization of cellulose microfibrils in the cell wall of TO-wood significantly decreased cell wall rigidity and improved the compressibility of the obtained xerogel (Fig. 4a). The compressive stress of TO-wood xerogel at 10% strain was 220 kPa, 55 times stronger than the highly mesoporous wood aerogel prepared from TO-wood by supercritical drying with $CO_2$[19]. The compression mainly eliminated the micrometer-sized lumina in the wood xerogel, as observed by FE-SEM (Fig. 4a). At a compressive strain of 80%, a high density (780 kg m$^{-3}$) xerogel sheet was obtained with a porosity of 48.1%.

Hence, the wood volume fraction in transparent wood could be controlled. Compression was applied perpendicular to the axial direction of native fiber cells in the TO-wood xerogel before the acrylic resin impregnation step (Fig. 1b). The TO-wood xerogels were compressed to different thicknesses to produce transparent wood with wood volume fractions up to 50%, which was also confirmed with the decomposition behavior of TO-wood/ABPE composites by thermogravimetric analysis (TGA), as shown in Supplementary Fig. 7.

Figure 4b shows the typical tensile stress-strain curves of the wood xerogel/ABPE composites along the fiber direction. From the uncompressed wood xerogels, both D-wood/ABPE (wood volume fraction of 17%) and TO-wood/ABPE composites (wood volume fraction of 12%) experienced mainly elastic deformation before failure. The strain-to-failure of the D-wood/ABPE composite was 1.1%, while the TO-wood/ABPE composite showed slightly lower extensibility with a strain-to-failure of 0.8% (Supplementary Table 3). The tensile strength and Young's modulus of the D-wood/ABPE composite were 88 MPa and 8.0 GPa, respectively. The TO-wood/ABPE composite showed a tensile strength of 59 MPa and a Young's modulus of 7.7 GPa.

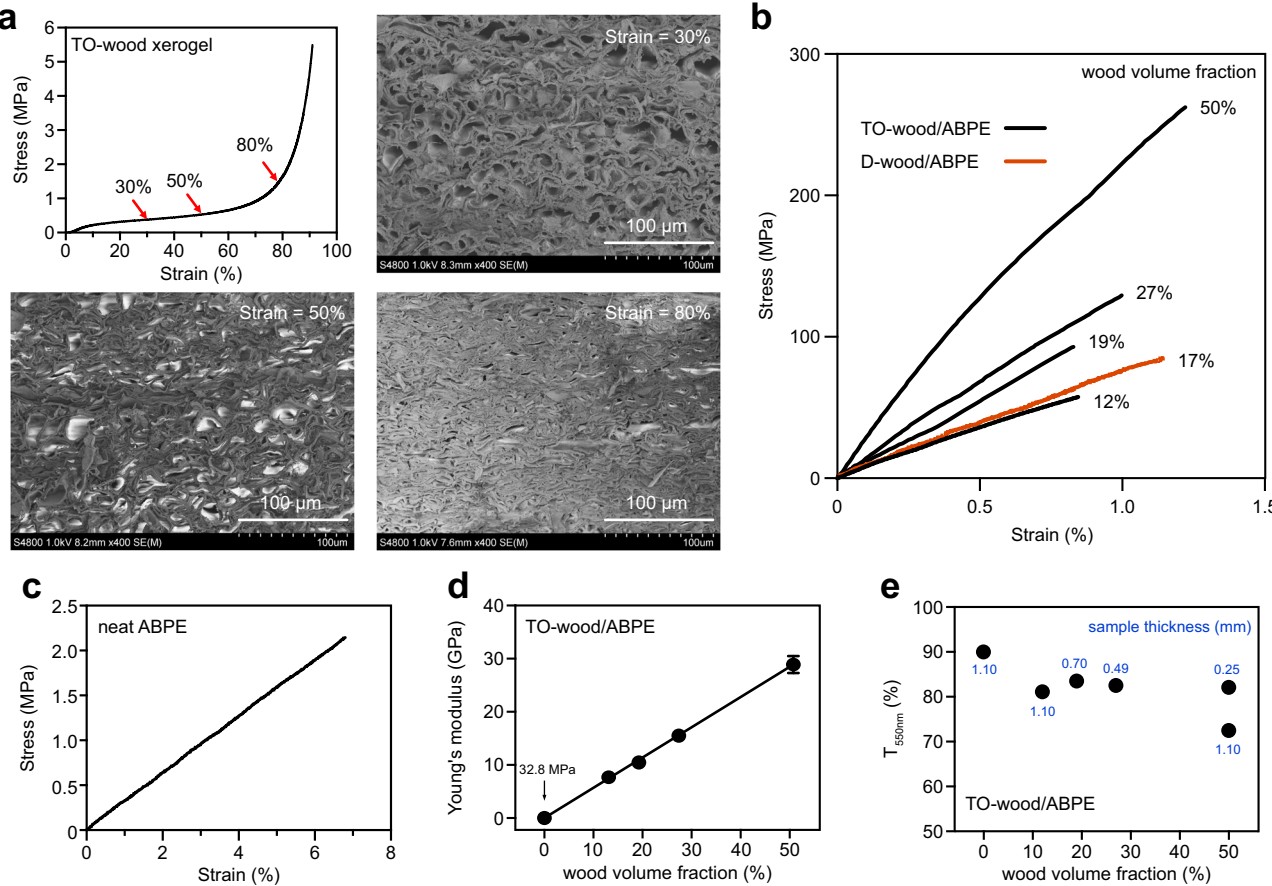

**Fig. 4 | Control wood volume fraction in transparent wood. a** Compressive stress-strain curve of the TO-wood xerogel with a density of 185 kg m⁻³ and FE-SEM micrographs showing the cross-sectional microstructure of the compressed TO-wood xerogels at compressive strains of 30%, 50%, and 80%, respectively. The red arrows indicate the xerogels obtained at different compressive strains for FE-SEM analysis and transparent wood preparation. **b** Typical stress-strain curves of the D-wood/ABPE and TO-wood/ABPE composites with different wood volume fractions under tensile deformation along the longitudinal direction. **c** Typical stress-strain curve of the neat ABPE acrylic resin sheet under tensile deformation. **d** Young's modulus of the TO-wood/ABPE composites as a function of the wood volume fraction. **e** Dependency of the total optical transmittance at 550 nm ($T_{550nm}$) of transparent wood on the wood volume fraction and sample thickness.

Compared to the neat ABPE acrylic resin sheet (Fig. 4c), which had a tensile strength of 2 MPa and a Young's modulus of 32.8 MPa, the reinforcing effect with the wood xerogels was remarkable. The mechanical performance of transparent wood composites was further improved by compressing the TO-wood xerogel before the resin impregnation. By increasing the wood volume fraction to 19%, the tensile strength and Young's modulus increased to 91 MPa and 10.5 GPa, respectively, exceeding the properties of the D-wood/ABPE composite. During strain hardening, both strain-to-failure and tensile strength increased simultaneously, which indicates the favored interfacial toughness between the TO-wood xerogel and ABPE matrix. On the tensile fracture surface, the composite reinforced with TO-wood showed a clean surface (Supplementary Fig. 8). By contrast, a severe amount of polymer pullout from the lumina was observed for the D-wood/ABPE composite. This distinct difference in deformation was suggested to originate from different extents of interfacial adhesion[35]. The acrylic resin polymer infiltrated in the intercellular spaces of the TO-wood cell walls formed a continuous phase interpenetrating with the TO-wood xerogel. It provided more efficient stress transfer under tensile deformation and a stiffened cell wall, with spontaneous failure of adjacent cell walls or polymer domains[35,36]. This was not seen for the D-wood/ABPE composite, as the middle lamella and cell corner remained densely packed and unopened. Furthermore, the transparent TO-wood/ABPE composite with a wood volume fraction of 50% showed a high tensile strength of 259 MPa and an exceptionally high Young's modulus of 29.0 GPa. The increase in Young's modulus

follows the rule of mixtures and showed a linear increase with increasing wood volume fraction (Fig. 4d), suggesting the feasibility of manipulating the mechanical properties of transparent wood by controlling the xerogel compression procedure.

The $T_{550nm}$ of the transparent TO-wood/ABPE composites remained constant at above 80% even at high wood volume fractions (Fig. 4e), suggesting that aggregation of cellulose microfibrils was not induced by compression of the TO-wood xerogel. Such high optical transmittance is remarkable, as it was reported previously that the $T_{550nm}$ of transparent wood decreased significantly from 85.0 to 34.6% when the wood volume fraction increased from 5 to 65%[2]. The haze value of transparent wood even decreased from 58.5% to 40.0% as the wood volume fraction increased from 12% to 50% since the thickness of the samples decreased from 1.10 mm to 0.25 mm (Supplementary Table 2). Thus, the transparent wood with 50% TO-wood xerogel with a thickness of 1.10 mm was also prepared and showed $T_{550nm}$ of 72.3% and haze of 73.9%. As comparison, by using high-density (593 kg m⁻³) wood species, birch wood volume fraction as high as 26% has been obtained for transparent wood with a $T_{550nm}$ of 87% and haze of 46% at a thickness of 1 mm (Supplementary Table 4)[37]. A wood volume fraction of 30% has also been achieved by using lignin-modified balsa wood to form transparent wood with $T_{550nm}$ of 90% and haze of 60% at a thickness of 1 mm[7]. However, both methods used solvent and vacuum-assisted resin impregnation and did not fabricate transparent wood with even higher wood volume fraction. Therefore, the wood xerogel compression procedure for fabricating transparent wood with

a wood volume fraction as high as 50% is important from both material efficiency, and high mechanical and optical performance perspectives.

To keep high wood volume fraction at 50% and demonstrate the scalability of wood xerogel, transparent wood with thickness of 6.05 mm was prepared (Fig. 1b). The haze of this transparent wood was significantly higher than the thinner samples as observed on a black background (Supplementary Fig. 9). The thick sample showed good optical transparency and anisotropic light scattering within the sample as illustrated by using a laser pointer (Supplementary Fig. 10). Further, transparent wood samples with large size (300 mm × 100 mm) were also prepared. (Supplementary Fig. 11).

## Discussion

Due to the highly porous nature of wood, transparent wood usually contains less than 10% wood mass, leaving the major component to be the polymer matrix[2,3]. Here, transparent wood with high wood volume fractions were successfully fabricated from the mesoporous wood xerogel impregnated with acrylic resin. The in situ individualized cellulose microfibrils in the cell wall of TO-wood were stabilized by crosslinking with trivalent aluminum ions. Replacement of water with the nonpolar solvent hexane allowed evaporative drying at ambient pressure to produce a xerogel maintaining the mesoporous cell wall structure and high specific surface area of $260\ m^2\ g^{-1}$. The mesoporous TO-wood xerogel was compressed to different thicknesses to prepare transparent wood with wood volume fraction in the range of 12–50%. Transparent wood with a wood volume fraction of 50% showed a total optical transmittance of 80% at 550 nm, and tensile strength and Young's modulus as high as 259 MPa and 29 GPa, higher than those of transparent wood previously reported in the literature (Supplementary Table 4). The wood xerogel demonstrated material efficiency in fabricating high-performance transparent wood. Its high specific surface area and high mesopore volume also hold great potential in applications spanning from insulation, filtration, and nanofluidic ion regulation to flexible devices.

## Methods

### Materials and chemicals

Balsa wood (*Ochroma pyramidale*) with oven dry densities of 190, 280, and 350 kg m$^{-3}$ was purchased from Material AB, Stockholm. Sodium chlorite, sodium hyperchloride, TEMPO, aluminum (III) chloride anhydrous, hexane (HPLC plus grade), ethanol (99.9%), and 2,2-dimethoxy-2-phenylacetophenone were purchased from Merck and used as received. Bisphenol A ethoxylate diacrylate ABPE-10 (refractive index = 1.536)[38] was kindly provided by Shin-Nakamura Chemical Co. Ltd., Japan.

### Preparation of TEMPO-oxidized wood

Balsa wood with thicknesses of 1, 5, and 25 mm in the tangential direction were cut into samples with dimensions of 50 mm × 20 mm, 50 mm × 40 mm, and 300 mm × 100 mm in longitudinal and radial directions by using a tabletop circular saw. Balsa wood samples were delignified in 1 wt.% NaClO$_2$ at pH 4.6 for 12–24 h depending on the thickness. The delignified wood samples were thoroughly washed with deionized water to remove any residual chemicals. The delignified wood samples were then oxidized in a NaClO$_2$/NaClO/TEMPO system[39]. The dry wood samples (1 g) were soaked in 100 ml sodium phosphate buffer (pH 6.8) dissolving 0.032 g TEMPO and 2.26 g NaClO$_2$. Then, 0.1 M NaClO in 20 ml sodium phosphate buffer (pH 6.8) was added. The reaction was carried out at 60 °C for 48 h. The TEMPO-oxidized wood samples were then immersed in 0.1 M AlCl$_3$ solution for 1 hour and then in 0.01 M AlCl$_3$ for 24 h for crosslinking. The Al$^{3+}$-crosslinked TO-wood sample was thoroughly washed with deionized water.

### Preparation of wood xerogel

The wet Al$^{3+}$-crosslinked TO-wood (ca. 0.5 g dry mass) was placed in a crystallization tank for solvent exchange. It was first immersed in

100 ml absolute ethanol for two days and then 100 ml in 99.5% hexane for two days. During each step, solvent was refreshed once after 24 h. The hexane exchanged TO-wood was sandwiched between two polytetrafluoroethylene (PTFE) membranes (pore size: 0.1 μm) and placed in a crystallization tank containing 100 ml hexane. The setup was then placed in a sealed desiccator with 500 g silica gels and the drying process was carried out at 60 °C in an oven for 48 h. During the drying, 94% of the hexane was adsorbed into the silica gels (Supplementary Fig. 12). The wood xerogel was obtained after cooling to room temperature. Before further characterization, the wood xerogel was conditioned at 50% RH and 23 °C. The D-wood xerogel was also prepared through the same solvent exchange-drying procedure as a comparison.

### Fabrication of transparent wood composites

Xerogels were immersed in a glass Petri dish containing the ABPE-10 monomer and 1 wt.% photoinitiator 2,2-dimethoxy-2-phenylacetophenone. The Petri dish was placed in the dark overnight under ambient conditions for the impregnation. The impregnated xerogels were sandwiched between borosilicate glass plates with spacers. A UV lamp (365 nm) was then used to cure the composites for 15 min. To prepare the TO-wood/ABPE composites with different wood volume fractions, the TO-wood xerogels were compressed with spacers to control the thickness before the resin impregnation and UV curing.

### Characterizations

Nitrogen adsorption/desorption isotherms were recorded on 3Flex adsorption analyzer (Micromeritics Instrument Corp., USA). The specific surface area was obtained by using Brunauer-Emmett-Teller theory. Field emission-scanning electron microscopy (FE-SEM) was performed on a Hitachi model S-4800 SEM (Japan) operating at a working distance of 8 mm and a voltage of 1 kV. Both tensile and compression tests were conducted on a universal tester Instron 5944 (MA, USA) equipped with a 2 kN load cell and a video extensometer. For the tensile test, samples with dimensions of 50 mm × 3 mm (longitudinal and radial directions) were stretched at a strain rate of 10% min$^{-1}$ in the longitudinal direction. A compression test was conducted on the TO-wood xerogel sample with a thickness of 5 mm in the tangential direction at a strain rate of 5% min$^{-1}$. The total transmittance and haze of the neat ABPE and composites in the range of 400–800 nm were measured with a spectrophotometer coupled with an integration sphere at 25 °C, according to ASTM D1003 Standard Method for Haze and Luminous Transmittance of Transparent Plastics. A quartz tungsten halogen light (model 66181, Oriel Instruments) was used as the incident beam. Each sample was measured at 3 different sample positions. SAXS measurements were conducted on a point collimated Anton Paar's SAXS point 2.0 system equipped with a microfocus X-ray source (Cu Kα radiation, wavelength 1.5418 Å, beam size of 500 μm), and an Eiger R 1 M Tilt detector with a pixel size of 75 × 75 μm$^2$. All measurements were performed at room temperature with a beam path pressure of approximately 1–2 mbar. The sample-to-detector distance was set to 576 mm. The exposure time of each measurement was set at 10 min. Prior to measurement, the composites were cut into matchstick dimensions and mounted with the beam perpendicular to the fiber direction of the samples. The data processing was performed by the SAXS analysis package (Anton Paar, Graz, Austria). The thermal properties of the composites were analyzed on a Mettler Toledo TGA/DSC1 (Switzerland). Approximately 5 mg of each sample was placed in an alumina crucible, stabilized at 110 °C for 10 min to remove moisture, and analyzed from 110 to 600 °C at a heating rate of 10 °C min$^{-1}$ under a nitrogen flow rate of 50 ml min$^{-1}$. The mass content of aluminum in the xerogel was characterized by inductively coupled plasma atomic emission spectroscopy (Thermo Scientific iCAP 600 series). The contact angle (CA) of a 3 μl droplet of ABPE resin on the surface of TO-wood xerogel and D-wood xerogel was measured under conditions of

25 °C and 50% relative humidity by a KSV instrument CAM 200 equipped with a Basler A602f camera.

## Data availability

The datasets generated and/or analyzed during the current study are supplied in the supplementary information. If additional data or information is sought, this will be provided by the corresponding author upon request. Source data are provided with this paper.

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

## Acknowledgements

This work is supported by the Knut and Alice Wallenberg Foundation (KAW) through the Wallenberg Wood Science Center (grant number:

WWSC 2.0: KAW 2018.0452) and the KAW Biocomposites project (grant number: 2018.0451 LAB).

## Author contributions

Q.Z., L.B., and S.W. conceived the project. Q.Z. and L.B. supervised the project. S.W. prepared samples and conducted gas adsorption analysis, SEM, mechanical test, and thermal analysis. L.L. performed X-ray scattering characterization. L.Z. performed optical measurement and contact angle measurement. S.K. contributed to sample preparation and illustration. Q.Z. and S.W. wrote the manuscript. All authors have read and agreed to the published version of the manuscript.

## Funding

## Competing interests

The authors declare no competing interests.
