## [Peer Review File · Nature Communications]

Wood Xerogel for Fabrication of High-Performance Transparent WoodReviewers' comments:

Reviewer #1 (Remarks to the Author):

The paper describes some new results in terms of high-performance transparent wood using wood xerogel as skeleton neither interface modification no vacuum infiltration. The obtained transparent wood shows both high optical transparency and high mechanical performance. The paper is interesting and published with minor revision. However, the following comments should be addressed prior to acceptance:

1. What's the advantage of wood xerogel used as skeleton compared with delignified wood? As we known, besides delignified wood, nanowood (Li et al, Science Advances, 2018, 4, eaar3724), wood sponge (Sun et al., ACS Nano, 2020, 14, 14665), wood aerogel (Garemark, ACS Appl. Mater. Interfaces, 2022, 14, 24697) also have been reported. Do these wood skeletons suitable for transparent wood without vacuum infiltration?
2. In order to elimination of interface debonding gap between the cell walls of wood skeleton and infiltrated polymer, both acetylation modification (Li et al., 2018, J. Mater. Chem. A, 2018, 6, 1094) and super low remaining lignin strategy (Li et al., 2019, J. Mater. Res., 34, 932) were developed to prepare thick, high performance transparent wood under vacuum infiltration of polymer. It should be cited these articles in introduction. What's the infiltration mechanism under ambient conditions? Strong capillary force generated by mesopores with sizes in the range of 2-50 nm (Line 44-45), or ionic crosslinking of carboxyl groups with Al^{3+} ?
3. Because there is some apparent gaps in the cell corner (Fig. 3d), it is better to take the photographs (Fig. 1d, 1h) of transparent wood on black substrate to be easily distinguish transparency performance (Li et al., 2019, J. Mater. Res., 34, 932). Please provide wider optical properties in the range of 400-900 nm (Fig. 3a-3b) since 400-500 nm is important to the optical performance of transparent wood.
4. Line 159, It should be Supplementary Fig.2 instead of Fig. 1. Fig 2a-2h, the scale is not clear.
5. The style of References should be consistent (22#, 27#-29#, etc.).

Reviewer #2 (Remarks to the Author):

The manuscript of Wang et al. describes a new method, based on wood xerogel formation, for the preparation of transparent wood.

The manuscript is well-written and the work described of great interest for researchers especially in the wood and cellulose materials as well as bio-based composites fields. However, in its current form it lacks enough originality and impact. The most significant advance of this work is represented, rather than the

final product (transparent wood), by the technique described that allows make a highly porous wood xerogel. However, this technique is basically the adaptation to wood of a procedure reported in ref. 21. The only original addition seems to be the crosslinking of microfibrils by aluminum ions, which as such would require more attention. I suggest that more details should also be added in the description of the technique itself for the preparation of wood xerogels. For example, aluminum ions are used as crosslinkers: it would be interesting to know how much of aluminium ions are actually required i.e. remain in the wood scaffold (especially in terms of environmental friendliness and possible applications). The xerogel is obtained from the controlled evaporation of hexane, but it is not clear how this happens: is the hexane being absorbed by the silica gel? The authors should provide more details to ensure proper reproducibility. What would happen if the dry xerogel would be put back into water, instead of being impregnated with resin? The procedure described is comprised of many different steps, which makes it questionable whether such an effort would be reasonable to obtain a form of transparent wood, which properties seem to be consistent with (that is, not especially superior to those of) previously reported examples.

There are currently several reports on transparent wood in the literature, but only modest improvements regarding its fabrication on application-relevant scales. In this sense, it is not evident from the reported results if the size samples reported are the upper limit, or if the technique described could allow even bigger (and maybe thicker) samples to be prepared. If the authors choose to keep the focus on transparent wood, then I would encourage to make a more detailed comparison with previously reported materials, not only as a function of the mechanical properties (as in Supplementary Table 4) but also on the optical ones (such as haze; showing a picture in this sense would also be helpful). I suggest that more details should also be added in the description of the technique itself for the preparation of wood xerogels.

The authors described results obtained only with balsa wood. Although I understand that balsa wood is a standard wood for this kind of experiments, it would be really interesting and relevant for practical applications to check the suitability of the proposed technique for other wood species as well.

In Figure 1, the schemes c,d,e are not sufficiently informative and should be redrawn/redesigned.

Response to the comments

We thank the reviewers for the excellent comments and feedback which helped us to improve the manuscript. We have made all possible efforts to address the comments and revised the manuscript accordingly. The changes made in the manuscript are highlighted in red. A point-by-point response to the comments by the reviewers is provided below.

Reviewer 1

The paper describes some new results in terms of high-performance transparent wood using wood xerogel as skeleton neither interface modification no vacuum infiltration. The obtained transparent wood shows both high optical transparency and high mechanical performance. The paper is interesting and published with minor revision. However, the following comments should be addressed prior to acceptance:

1. What's the advantage of wood xerogel used as skeleton compared with delignified wood? As we known, besides delignified wood, nanowood (Li et al, Science Advances, 2018, 4, eaar3724), wood sponge (Sun et al., ACS Nano, 2020, 14, 14665), wood aerogel (Garemark, ACS Appl. Mater. Interfaces, 2022, 14, 24697) also have been reported. Do these wood skeletons suitable for transparent wood without vacuum infiltration?

Response:

The advantage of using TO-wood xerogel over delignified wood is solvent and vacuum free infiltration of polymers into the fiber cell walls enabled by the high mesoporosity presented in TO-wood. The delignification of wood is generally achieved by either the pulping process, or the bleaching process, or a combination of both. The most used pulping chemistry for wood delignification employs nucleophile SO_3^{2-} in alkaline condition, usually with NaOH and Na_2SO_3 . Nanowood (Li et al, Science Advances, 2018, 4, eaar3724) was produced in NaOH/ Na_2SO_3 system and an additional bleaching process with hydrogen peroxide was used. Dry Nanowood was obtained by freeze drying. Although specific surface area (S_{BET}) data was not reported for Nanowood, the S_{BET} of delignified wood produced by using the same pulping process was reported to be $13.8 \text{ m}^2/\text{g}$ (Song et al. ACS Nano 2018, 12, 1, 140–147), which is lower than the D-wood xerogel ($37 \text{ m}^2/\text{g}$) reported in this work. Wood sponge (Sun et al., ACS Nano, 2020, 14, 14665) was prepared by bleaching wood with a mixture of hydrogen peroxide and acetic acid, followed by freeze drying. The S_{BET} of delignified wood prepared in a similar process was $20 \text{ m}^2/\text{g}$ (Fu et al., ACS Appl. Mater. Interfaces, 2017, 9, 41, 36154–36163). Therefore, the permeability of fiber cell walls of Nanowood and wood sponge is not expected to be better than the D-wood xerogel reported in this work, as infiltrated polymer resin was mainly found inside the lumina. Wood aerogel (Garemark, ACS Appl. Mater. Interfaces, 2022, 14, 24697) was prepared by using ionic liquid to partially dissolve and regenerate cellulose in delignified wood.

It was reported to have high S_{BET} of 181 m^2/g when supercritical drying was used. However, the regenerated cellulose nanofibrous network structure is mainly in the lumina of wood fiber cells. The mesoporosity of the fiber cell walls is not increased as compared to delignified wood. So this wood aerogel is also not suitable for transparent wood without vacuum infiltration.

We have integrated the above discussion in the Introduction as shown below:

Removal or partial removal of lignin usually generates mesopores which endows delignified wood with S_{BET} in the range of 9–41 $\text{m}^2 \text{g}^{-1}$ depending on the drying methods¹⁵. The delignification of wood is generally achieved by either pulping process, or bleaching process, or a combination of both⁴. Nanowood was produced by the alkaline sulfite pulping process using $\text{NaOH}/\text{Na}_2\text{SO}_3$, followed by an additional bleaching process with H_2O_2 ¹⁶. Such delignified wood had a S_{BET} of 13.8 $\text{m}^2 \text{g}^{-1}$ after freeze drying¹⁷. Wood sponge prepared by the bleaching process with a mixture of H_2O_2 and acetic acid had a S_{BET} of 20 $\text{m}^2 \text{g}^{-1}$ after freeze drying^{15, 18}. Delignified wood aerogel with a S_{BET} of 36 $\text{m}^2 \text{g}^{-1}$ was also produced by the bleaching process with acidified sodium chlorite followed by supercritical drying with CO_2 ¹⁹

2. In order to elimination of interface debonding gap between the cell walls of wood skeleton and infiltrated polymer, both acetylation modification (Li et al., 2018, J. Mater. Chem. A, 2018, 6, 1094) and super low remaining lignin strategy (Li et al., 2019, J. Mater. Res., 34, 932) were developed to prepare thick, high performance transparent wood under vacuum infiltration of polymer. It should be cited these articles in introduction. What's the infiltration mechanism under ambient conditions? Strong capillary force generated by mesopores with sizes in the range of 2-50 nm (Line 44-45), or ionic crosslinking of carboxyl groups with Al^{3+} ?

Response:

Indeed, interface modification is a very important strategy to eliminate the interface debonding gap between the cell walls of wood skeleton and infiltrated polymer. We have added the discussion and cited the articles in the Introduction as shown below:

Further chemical modification of delignified wood by surface acetylation¹¹ or elimination of excessive lignin¹² allowed the preparation of thick yet transparent wood composites.

The fast infiltration of the resin under ambient conditions is driven by the high mesopore volume of 0.37 $\text{cm}^3 \text{g}^{-1}$ and strong capillary force generated in the slit-like pore geometry with pore sizes in the range of 2-20 nm inside the cell wall of TO-wood xerogel (Fig. 2j). In addition, the accessible intercellular space was also found as shown in Fig. 2d. We have also tested the infiltration of the resin in non-crosslinked TO-wood xerogel under ambient conditions, which was good as well. However, the strength of the non-crosslinked TO-wood xerogel was much

weaker than the Al^{3+} -crosslinked TO-wood xerogel. We have added the results of contact angle measurement and tensile test in the revised manuscript as shown below:

The fast infiltration of the resin in TO-wood xerogel was also confirmed by contact angle measurement (Supplementary Fig. 5). The ABPE resin had an initial contact angle of 73.8° on the TO-wood xerogel surface and the droplet was quickly adsorbed into the wood within 5 s. By contrast, the resin droplet on D-wood xerogel had an initial contact angle of 92.5° which decreased to 60.8° at 5 s and remained at 58.4° after 10 minutes.

Supplementary Fig. 5. Contact angle test of ABPE resin droplets ($3 \mu\text{l}$) on the surfaces of TO-wood and D-wood xerogels. The infiltration of ABPE resin monomer into xerogel surface was much faster for the TO-wood as the droplet was already absorbed into TO-wood xerogel after 5 s. By contrast, the droplet remained on the D-wood xerogel surface with a contact angle of 58.4° after 10 min.

The non-crosslinked sample showed tensile strength (σ) of 0.80 ± 0.16 MPa, Young's modulus (E) of 0.11 ± 0.01 GPa, and strain-to-failure (ε) of 1.36 ± 0.12 % along the longitudinal direction (Supplementary Fig. 2). After crosslinking with Al^{3+} ions, the mechanical properties of TO-wood xerogel were significantly improved, showing σ of 4.81 ± 0.19 MPa, E of 0.32 ± 0.04 GPa, and ε of 3.62 ± 0.96 %. This indicates that the crosslinking with Al^{3+} ions reduced the structural defects in the wood xerogel through enhanced interfibrillar bonding of cellulose microfibrils.

Supplementary Fig. 2. Tensile mechanical stress-strain curves of the Al^{3+} -crosslinked and non-crosslinked TO-wood xerogels.

3. Because there are some apparent gaps in the cell corner (Fig. 3d), it is better to take the photographs (Fig. 1d, 1h) of transparent wood on black substrate to be easily distinguish transparency performance (Li et al., 2019, *J. Mater. Res.*, 34, 932). Please provide wider optical properties in the range of 400-900 nm (Fig. 3a-3b) since 400-500 nm is important to the optical performance of transparent wood.

Response:

We took photos of transparent wood samples on a black substrate. These photos were added in the Supplementary Information as Fig. 9 as shown below.

We agree that the range of 400–500 nm provides useful information on the optical performance of transparent wood. We performed optical transmittance and haze measurements using a new light source which can provide stable reading in range of 400-500 nm. However, with the new setup, the transmittance above 800 nm was not measurable. 400–800 nm is a commonly used range for characterizing the optical properties of transparent wood (Zhu et al. *Adv. Mater.* 28, 26, 2016). Therefore, we updated transmittance and haze plots within the range of 400-800 nm in Fig.3a and 3b in main text and as shown below. The transmittance and haze values in Supplementary Table 3 were also updated.

To keep high wood volume fraction at 50% and demonstrate the scalability of wood xerogel, transparent wood with thickness of 6.05 mm was prepared (Fig. 1b). The haze of this transparent wood was significantly higher than the thinner samples as observed on a black background (Supplementary Fig. 9).

Supplementary Fig. 9. Photographs of the control D-wood/ABPE composite and the transparent TO-wood/ABPE composites of different thickness on black background.

Fig. 3 Optical properties and structural characterization of transparent wood.

4. Line 159, It should be Supplementary Fig.2 instead of Fig. 1. Fig 2a-2h, the scale is not clear.

Response:

The citing of Supplementary Fig.2 was corrected, and the scale bars were added for the SEM images in Fig. 2, Fig. 3, and Fig. 4.

5. The style of References should be consistent (22#, 27#-29#, etc.).

Response:

We have corrected the styles of references according to the guideline of Nature Communications.

Reviewer 2

The manuscript of Wang et al. describes a new method, based on wood xerogel formation, for the preparation of transparent wood. The manuscript is well-written and the work described of great interest for researchers especially in the wood and cellulose materials as well as bio-based composites fields.

Response:

The use of mesoporous wood xerogel for the preparation of transparent wood with high wood volume fraction and high mechanical performance without compromising the optical transmittance has never been reported previously. We thank the reviewer for highlighting the novelty of our work.

However, in its current form it lacks enough originality and impact. The most significant advance of this work is represented, rather than the final product (transparent wood), by the technique described that allows make a highly porous wood xerogel.

Response:

The concept of transparent wood has been developed since 2016 (Li et al. *Biomacromolecules* 2016, 17, 4, 1358–1364 and Zhu et al. *Adv. Mater.* 2016, 28, 26, 5181–5187). **Due to the highly porous nature of wood, transparent wood usually contains less than 10% wood mass, leaving the major component to be the polymer matrix^{2,3}. Here, transparent wood with high wood volume fractions** were successfully fabricated from the mesoporous wood xerogel impregnated

with acrylic resin. The in situ individualized cellulose microfibrils in the cell wall of TO-wood were stabilized by crosslinking with trivalent aluminum ions. Replacement of water with the nonpolar solvent hexane allowed evaporative drying at ambient pressure to produce a xerogel maintaining the mesoporous cell wall structure and high specific surface area of $260 \text{ m}^2 \text{ g}^{-1}$. The mesoporous TO-wood xerogel was compressed to different thicknesses to prepare transparent wood with V_f in the range of 12–50%. **Transparent wood with a V_f of 50% showed an optical transmittance of 80% at 550 nm, and tensile strength and Young's modulus as high as 259 MPa and 29 GPa, higher than those of transparent wood previously reported in the literature (Supplementary Table 4).** The wood xerogel demonstrated material efficiency in fabricating high-performance transparent wood. Its high specific surface area and high mesopore volume also hold great potential in applications spanning from insulation, filtration, and nanofluidic ion regulation to flexible devices.

However, this technique is basically the adaptation to wood of a procedure reported in ref. 21. The only original addition seems to be the crosslinking of microfibrils by aluminum ions, which as such would require more attention. I suggest that more details should also be added in the description of the technique itself for the preparation of wood xerogels. For example, aluminum ions are used as crosslinkers: it would be interesting to know how much of aluminium ions are actually required i.e. remain in the wood scaffold (especially in terms of environmental friendliness and possible applications).

Response:

The procedure for wood xerogel preparation was adapted from a previously reported method (Sakuma et al. *ACS Nano*, 2021, 15, 1436–1444), in which the use of hexane to preserve the mesoporosity of nanocellulose network was proposed. This inspired us to use hexane to prepare highly mesoporous xerogel from wood for the first time in literature. Xerogel has been investigated since it allows the preparation of mesoporous aerogels without using energy intensive supercritical drying or freeze drying. Silica xerogels (Müller et al. *Adv. Mater.* 2000, 12, 22), nanocellulose xerogels (Toivonen et al. *Adv. Func. Mater.* 2015, 25, 42, and Li et al. *Chem. Eng. J.* 2019, 366, 531–538), MOFs xerogels (Saha et al. 2014, 136, 42) have already been reported. TEMPO-mediated oxidation of delignified wood (Li et al. *Adv. Mater.* 2020, 32, 2003653) and subsequent crosslinking with aluminum ions (Wang et al. *ACS Appl. Mater. Interfaces*, 2021, 13, 25, 29949–29959) were adapted from our previous work. These were already mentioned and cited in our manuscript.

Following the reviewer's suggestion, we quantified the aluminum content in our wood xerogel by using ICP-OES. The aluminum content was 0.8 wt.% in the wood xerogel. Therefore, the aluminum content in the final transparent wood was up to 0.4 wt.% as the wood volume fraction was up to 50%. We also added detailed description of xerogel preparation in the Methods section

as shown below. The effect of crosslinking with Al^{3+} ions on the tensile mechanical properties of TO-wood xerogel was also characterized (check the reply to the question 2 by reviewer 1).

The xerogel preparation procedure was adapted from a previously reported method²⁵. The wet Al^{3+} -crosslinked TO-wood (ca. 0.5 g dry mass) was placed in a crystallization tank for solvent exchange. It was first immersed in 100 ml absolute ethanol for two days and then 100 ml in 99.5% hexane for two days. During each step, solvent was refreshed once after 24 hours.

The xerogel is obtained from the controlled evaporation of hexane, but it is not clear how this happens: is the hexane being absorbed by the silica gel? The authors should provide more details to ensure proper reproducibility.

Response:

Regarding the collection of hexane, industrially, n-hexane is used as extraction solvent for vegetable oil and fish oil manufacturing at large scale (Survey of n-hexane, Mikkelsen, S., Warming, M., Syska, J., Voskian, A., Danish Ministry of the Environment, 2014). Hexane is commonly recovered and recycled to reduce the environmental impact and to cut cost. Hexane is highly volatile and therefore can be recovered by distillation and condensation. Owing to its low boiling point (69 °C) and immiscibility with water, hexane can be easily separated from mixture with various solvents, such as toluene (110 °C) and ethanol (78°C). Industrial scale system for recovery of hexane from heterogeneous system is also commercially available: <https://www.sulzer.com/en/shared/products/solvent-recovery-systems>.

To verify the collection of hexane by silica gels, the weight change of silica gels before and after the drying of wood xerogel was measured. The result showed that 94% of the added 100 ml hexane was adsorbed to silica gel, measured by a weight increase of 61.8 g, which is equivalent to 94.35 ml hexane taking the density of hexane as 655 kg m^{-3} . We have added the photographs of weight change of silica gels as Supplementary Fig. 12. We have also included related description in the experimental section. It is shown as following:

The hexane exchanged TO-wood was sandwiched between two PTFE membranes (0.1 μm) and placed in a crystallization dish containing 100 ml hexane. The setup was then placed in a sealed desiccator with 500 g silica gel and the drying process was carried out at 60 °C in an oven for 48 hours. During the drying, 94% of the hexane was adsorbed into the silica gels (Supplementary Fig. 12).

Supplementary Fig. 12. The weight change of silica gel before and after the drying of wood xerogel with 100 mL hexane ($\rho = 0.655 \text{ g cm}^{-3}$).

What would happen if the dry xerogel would be put back into water, instead of being impregnated with resin?

Response:

We think this question is not relevant as the TO-wood has been solvent exchanged from water to hexane before drying in order to achieve better infiltration of the hydrophobic resin under ambient conditions. Nonetheless, out of curiosity, we put the dry xerogel back into the water. As shown in the Figure below, the center of the xerogel was still white (incomplete swelling) after soaking in water for 4 days at ambient conditions, indicating that the water cannot infiltrate inside the cell wall.

Figure. Thickness and weight changes of dry TO-wood xerogel in water after 96 h.

The procedure described is comprised of many different steps, which makes it questionable whether such an effort would be reasonable to obtain a form of transparent wood, which properties seem to be consistent with (that is, not especially superior to those of) previously reported examples.

The wood xerogel preparation involved two steps: wood modification (delignification and oxidation) and drying with hexane. The infiltration of resin was performed under ambient conditions. As indicated by reviewer #1, such preparation procedure for transparent wood is new as neither interface modification nor vacuum infiltration was needed. Particularly transparent wood with high wood volume fraction and high mechanical performance without compromising the optical transmittance was achieved.

For mechanical properties, our transparent wood with a V_f of 50% (thickness 1.10 mm) showed an optical transmittance of 80% at 550 nm, and σ and E as high as 259 MPa and 29 GPa, higher than those of transparent wood previously reported in the literature (Supplementary Table 4).

For optical properties, the T_{total} of the transparent TO-wood/ABPE composites remained constant at above 80% even at high wood xerogel volume fractions (Fig. 4h), suggesting that aggregation of cellulose microfibrils was not induced by compression of the TO-wood xerogel. Such high optical transmittance is remarkable, as it was reported previously that the T_{total} of transparent wood decreased significantly from 85.0 to 34.6% when the wood volume fraction increased from 5 to 65%². The haze value of transparent wood even decreased from 58.5% to 40.0% as the wood xerogel volume fraction increased from 12% to 50% since the thickness of the samples decreased from 1.10 mm to 0.25 mm (Supplementary Table 2). Thus, the transparent wood with 50% TO-wood xerogel with a thickness of 1.10 mm was also prepared and showed T_{total} of 72.3% and haze of 73.9%. As comparison, by using high-density wood species, birch ($\rho = 593 \text{ kg m}^{-3}$) wood template volume fraction as high as 26% has been obtained for transparent wood with a T_{total} of 87% and haze of 46% at a thickness of 1 mm (Supplementary Table 4)³⁷. A wood volume fraction of 30% has also been achieved by using lignin-modified balsa wood to form transparent wood with T_{total} of 90% and haze of 60% at a thickness of 1 mm⁷. However, both methods used solvent and vacuum-assisted resin impregnation and did not fabricate transparent wood with even higher wood volume fraction. Therefore, the wood xerogel compression procedure for fabricating transparent wood with a wood volume fraction as high as 50% is important from both material efficiency, and high mechanical and optical performance perspectives.

There are currently several reports on transparent wood in the literature, but only modest improvements regarding its fabrication on application-relevant scales. In this sense, it is not evident from the reported results if the size samples reported are the upper limit, or if the technique described could allow even bigger (and maybe thicker) samples to be prepared. If the authors choose to keep the focus on transparent wood, then I would encourage to make a

more detailed comparison with previously reported materials, not only as a function of the mechanical properties (as in Supplementary Table 4) but also on the optical ones (such as haze; showing a picture in this sense would also be helpful). I suggest that more details should also be added in the description of the technique itself for the preparation of wood xerogels.

Response:

We agree that fabrication on application-relevant scales is important for transparent wood preparation. Our process provided a feasible way to produce highly mesoporous wood xerogel which can be used directly for transparent wood production, demonstrating significant advance in the potential up-scaling of transparent wood production. The dimension of the transparent wood prepared in the initial submission was not the upper limit of such technique. As suggested, we have also added comparison of optical properties in Supplementary Table 4 and the related discussion was added in optical properties in the manuscript as shown in response to previous question.

The thick sample showed good optical transparency and anisotropic light scattering within the sample as illustrated by using a laser pointer (Supplementary Fig. 10). Further, transparent wood samples with large size (300 mm × 100 mm) were also prepared. (Supplementary Fig. 11).

Supplementary Fig. 10. Anisotropic light scattering behavior of the transparent TO-wood/ABPE composite (thickness: 6.05 mm, wood volume fraction: 50%) with incident green spotlight as compared to neat ABPE acrylic resin sheet (thickness: 6.0 mm). The diameter of the laser point is 1 mm. The arrows indicate the longitudinal direction of the fiber cells in transparent wood.

Supplementary Fig. 11. Preparation of TO-wood/ABPE transparent wood with dimension of 300 mm × 100 mm (longitudinal × radial directions).

Supplementary Table 4. Mechanical and optical properties of transparent wood in literatures.

Wood template	Polymer matrix	V_f	σ_L (MPa)	E_L (GPa)	T_{total}^{**} (%)	Haze (%)	Thickness (mm)
TO-wood xerogel	ABPE	50%	259	29.0	82.1	40.0	0.25
TO-wood xerogel	ABPE	50%	259	29.0	72.3	73.9	1.10
D-wood	PMMA	25%	262.7	19.3	70	70	0.65
D-wood	PMMA	N/A	45.92	2.66	80.6	N/A	0.5
D-wood	PMMA	19%	90.1	3.59	85.0	71.0	1.2
D-wood	PMMA	12%	62.5	4.3	90-95	50-60	0.8
D-wood	PMMA	N/A	60.1	2.67	86	90	5
D-wood	PEG/PMMA	25.8%	70.5	14.9	68	77	1.5
D-wood	PMMA	6.4%	41.4	4.8	70.6	76.3	2
D-wood	PLIMA	26%	146.6	12.6	87	46	1.2
D-wood	Epoxy	N/A	45.38	2.37	90	~100	3
D-wood	Epoxy	N/A	76.28	1.4	68.2	N/A	2.5
D-wood	Epoxy	2.5%	43.39	N/A	90	10	0.7
D-wood	Thiol-ene	N/A	61	3.6	66	47	1.1
D-wood	Thiol-ene	28%*	179	12.3	86	50	1.1
D-wood	Thiol-ene	6.8%	50.7	4.11	~85	N/A	1
D-wood	Thiol-ene	4.3%	59	3.4	90	36	1.2
D-wood	PVA	29%	39.9	1.51	80	90	1
D-wood	PVA	N/A	143	3.85	91	15	0.8
D-wood	Polyimide	N/A	169	2.11	75 _{750 nm}	75	0.2
D-wood	MF	25%	60	11.1	74	66	1.2
esterified D-wood	PLIMA	26%	173.6	17.3	90	30	1.2
partially delignified wood	PMMA	N/A	171.4	N/A	61 _{800 nm}	N/A	0.42
partially delignified wood	Epoxy	N/A	91.95	N/A	80 _{600 nm}	93	2
lignin modified wood	Epoxy	30%	46	N/A	90	60	1

*Only weight fraction data available; ** Value is obtained at 550 nm if not specified.

V_f : wood template volume fraction; σ_L : tensile strength along longitudinal direction;

E_L : Young's modulus along longitudinal direction.

PMMA: polymethyl methacrylate, PVA: polyvinyl alcohol, PLIMA: poly(limonene acrylate),

PEG: polyethylene glycol, MF: melamine formaldehyde.

The authors described results obtained only with balsa wood. Although I understand that balsa wood is a standard wood for this kind of experiments, it would be really interesting and relevant for practical applications to check the suitability of the proposed technique for other wood species as well.

Response:

We agree it is interesting to see how xerogel process works on the other wood species. We have prepared xerogel and respective transparent wood using birch. The result is presented below. The result indicates that it is possible to fabricate large dimension transparent wood by using TO-wood xerogel from birch.

Figure. Photographs of native birch wood, birch TO-wood xerogel and birch TO-wood/ABPE transparent wood composite with wood volume fraction of 17% and dimension of 300 mm × 100 mm × 0.7 mm (longitudinal × radial × tangential directions). Native birch wood has an oven dry density of 534 kg m⁻³.

In Figure 1, the schemes c,d,e are not sufficiently informative and should be redrawn/redesigned.

Response:

We have redesigned Fig. 1 and the new figure is shown as following:

Fig. 1 Fabrication of wood xerogels and transparent wood.

a Photographs showing the white D-wood/ABPE composite (wood volume fraction of 17%) and the transparent TO-wood/ABPE composite (wood volume fraction of 12%) prepared from the corresponding D-wood and TO-wood xerogels with arrows indicating the axial direction of fiber cells as in the native balsa wood. **b** Schematic illustration for compression of the TO-wood xerogel to prepare thick transparent wood with a wood volume fraction of 50% and corresponding photographs of the samples.

REVIEWERS' COMMENTS

Reviewer #1 (Remarks to the Author):

Although, the procedure described is comprised of several steps, the paper describes some new results in terms of high-performance transparent wood using wood xerogel as skeleton neither interface modification no vacuum infiltration. The obtained transparent wood shows both high optical transparency and high mechanical performance. It provide an alternative strategy to high-performance transparent wood, which is original and novelty enough to NC.

Reviewer #2 (Remarks to the Author):

The authors have properly amended the manuscript according to the suggestions.

Response to the comments

Reviewer #1 (Remarks to the Author):

Although, the procedure described is comprised of several steps, the paper describes some new results in terms of high-performance transparent wood using wood xerogel as skeleton neither interface modification no vacuum infiltration. The obtained transparent wood shows both high optical transparency and high mechanical performance. It provide an alternative strategy to high-performance transparent wood, which is original and novelty enough to NC.

Response:

Thank you.

Reviewer #2 (Remarks to the Author):

The authors have properly amended the manuscript according to the suggestions.

Response:

Thank you.